# Exploring Nonlinear Intra-Annual Growth Dynamics in *Fagus sylvatica* L. Trees at the Italian ICP-Forests Level II Network

**Carlotta Ferrara** [1], **Maurizio Marchi** [2,*], **Gianfranco Fabbio** [2], **Silvano Fares** [1], **Giada Bertini** [2], **Maurizio Piovosi** [2] and **Luca Salvati** [1]

[1] CREA–Research Centre for Forestry and Wood, Rome, Italy
[2] CREA–Research Centre for Forestry and Wood, Arezzo, Italy
* Correspondence: maurizio.marchi85@gmail.com; Tel.: +39-0575-353-021

**Abstract:** The European beech (*Fagus sylvatica* L.) is a widely distributed tree species across Europe, highly sensitive to climate change and global warming. This study illustrates results of a 5-year monitoring time period from eight sites of the ICP-Forests Level II (intensive monitoring network) along the Italian latitudinal gradient. The tree-level relationship between tree growth dynamics and environmental factors, including seasonal climate fluctuations were investigated by means of tree-level Generalized Additive Mixed Models (GAMMs). Model results revealed that climate was responsible for just a portion of the variability in beech growth dynamics. Even if climatic predictors were highly significant in almost all sites, the model explained nearly 30% of the total variance (with just a maximum value of 71.6%), leaving the remaining variance unexplained and likely connected with forest management trajectories applied to each site (e.g., aged coppice and fully grown high forest). Climate change scenarios were then applied to predict site-specific future responses. By applying climate change scenarios, it was predicted that central and northern Italy would face similar climatic conditions to those currently detected at southern latitudes. A special case study was represented by VEN1 plot (Veneto, Northern Italy) whose current and future climate regimes were grouped in a unique and separated cluster.

**Keywords:** forest monitoring; climate change; dendrometer; stem growth; climate change scenarios

## 1. Introduction

Forests are believed to provide a large set of ecosystem services in addition to wood and biomass production [1–3]. Climate change can reduce, or at least alter, quality and quantity of the ecosystem services that forests provide, thus affecting resilience and cultural identity of local communities [4–6]. It is, therefore, of primary importance to develop and use reliable and adequate climatic surfaces for the purpose [7–9]. This would impact on the possibility to derive useful instruments for policymakers able to anticipate changes in forest ecosystems' functionality and their capacity to deliver services [10]. Ecosystem services of forests are related to specific functions, such as growth and carbon stock into woody tissues, carbon sequestration and oxygen enrichment via photosynthesis, regeneration, element cycling, soil development, biodiversity holding, and so on. For this reason, a wide set of attributes should be taken into account to evaluate the overall functionality of forests and their capacity to provide ecosystem services [11,12].

The spatial distribution of forest tree species (range) is often recognized as a good proxy to derive their ecological requirements and to build up statistical models [9,13]. Analysis of field data and tree-level information allows a comprehensive understanding of stand-level health as well as the

impact of environmental factors on tree growth dynamics [14–17]. The growth rate (increment) of forest tree-species and their standing biomass is a fundamental indicator to evaluate their health and stability across space and time [18–20]. This aspect is also directly connected to carbon sequestration and climate regulation (regulatory services) and not only to the economic value of the forest (provisional services). Growth integrates the physiological condition of trees and the associated changes are a reliable indicator of climate impact [21]. Therefore, tree growth is a key variable evaluating the ability of forests to adapt to a changing climate, to mitigate climate change and to provide valuable ecosystem services [22].

Expected impacts of climate change on forest ecosystems' health include shifted tree species distributions, a lower growth rate and changes in species composition and related spatial arrangement. These processes are mainly associated with rising mean temperatures, variations in precipitation regimes, longer drought conditions and the increasing frequency of extreme weather events [6,23–26]. Europe-wide reduction in primary productivity caused by heat waves and droughts was already detected in 2003 [27]. More recently, a widespread tree mortality has been observed in forests around the globe [28,29]. Temperature extremes, precipitation extremes and wind throws have been pointed out as the most important climatic and atmospheric stressors for forest health [25,30].

Under climate change, the complexity of forest ecosystems demands the development of a prompt and effective system for detecting the impacts of pressure factors. In Europe, such activities are carried out routinely by means of terrestrial surveys [31,32] within the International Co-operative Program on 'Assessment and Monitoring of Air Pollution Effects on Forests' (ICP Forests), which is the most comprehensive European program for large-scale detection of forest ecosystem health [33,34]. An interdisciplinary research agenda integrated with monitoring networks and modelling is needed to provide information at all levels of decision making, from policy development to the management unit [25,35,36].

The European beech (*Fagus sylvatica* L.) is one of the most frequently assessed deciduous tree species within the ICP Forests monitoring program. This tree can be found on both Level I (extensive network) and Level II (intensive network) plots along the North-South and West-East gradients in Europe ranging from Southern Scandinavia to Southern Italy, and from the Atlantic coast of Northern Spain to the Bulgarian Black Sea coast. A total of 87 beech Level II plots wide spread over Europe are currently being monitored. The relevance of this species in the scientific literature is also well known, given its sensitivity to climatic fluctuations and spatial extent [37,38]. The impact of climate change on the geographic distribution of European beech was explored by means of dendrochronological studies [39,40], phenological methods [41], species distribution modelling techniques and process-based approaches [13,42].

Since the Italian peninsula represents the rear-edge for the species, many ecological studies were carried out in the last decades on beech forests. Elevation was significantly correlated with dendrochronological variables along the Alps and Northern Apennines and this correlation diverges in central-southern Apennines [43]. Summer drought was recognized to be the most influencing driver with different effects at different elevations [39,40]. Studies on old-growth forests revealed the roles of disturbance, competition and climate in structuring tree stands [39,44]. Taken together, earlier studies indicate that climate change will impact adaptive capacity and spatial structure of forests along both latitudinal and elevation gradients.

In this paper, a modelling analysis is proposed, with the aim to evaluate the intra-annual growth rate of European beech and identify possible climate-growth interactions. The species was monitored across eight sites distributed along the whole Italian peninsula. Eight 5-year inter-annual time series datasets on tree growth within the Italian ICP-Forests Level II intensive monitoring network were evaluated in correlation with local climatic variables. Statistical techniques were applied to evaluate the influence of the main climatic variables on beech growth trends, to assess whether ecological conditions affected local growth and where (which plots) a potential source of adaptation to climatic drivers (i.e.,

growth dynamics not significantly affected by climate variables), could be detected. Finally, different climate change scenarios were applied to predict future development at each site.

## 2. Materials and Methods

### 2.1. Spatial Distribution of ICP-Forests Monitoring Network for European Beech in Italy

The European Beech is the most frequent deciduous tree species across the whole European ICP-Forests Level II network. Differing from Level I (i.e., the extensive monitoring network), Level II sampling design followed a non-probabilistic scheme and includes a wide set of sites and stand variables to be measured. Over the last decade, due to the increased survey costs and the reduced technical staff, a lower number of plots have been gradually monitored over time [32]. Despite this, beech forest plots' set remained unchanged in Italy (Figure 1) and climate types covered by the current network spans from mountain Mediterranean sites with higher mean temperatures and low precipitation in the growing season (CAL1, CAM1, PUG1), to damp, inner areas of Southern to Central and Northern Apennines with a cold winter season (ABR1, TOS3, LIG1), up to the sites located in Western (PIE1) and Eastern Alps (VEN1).

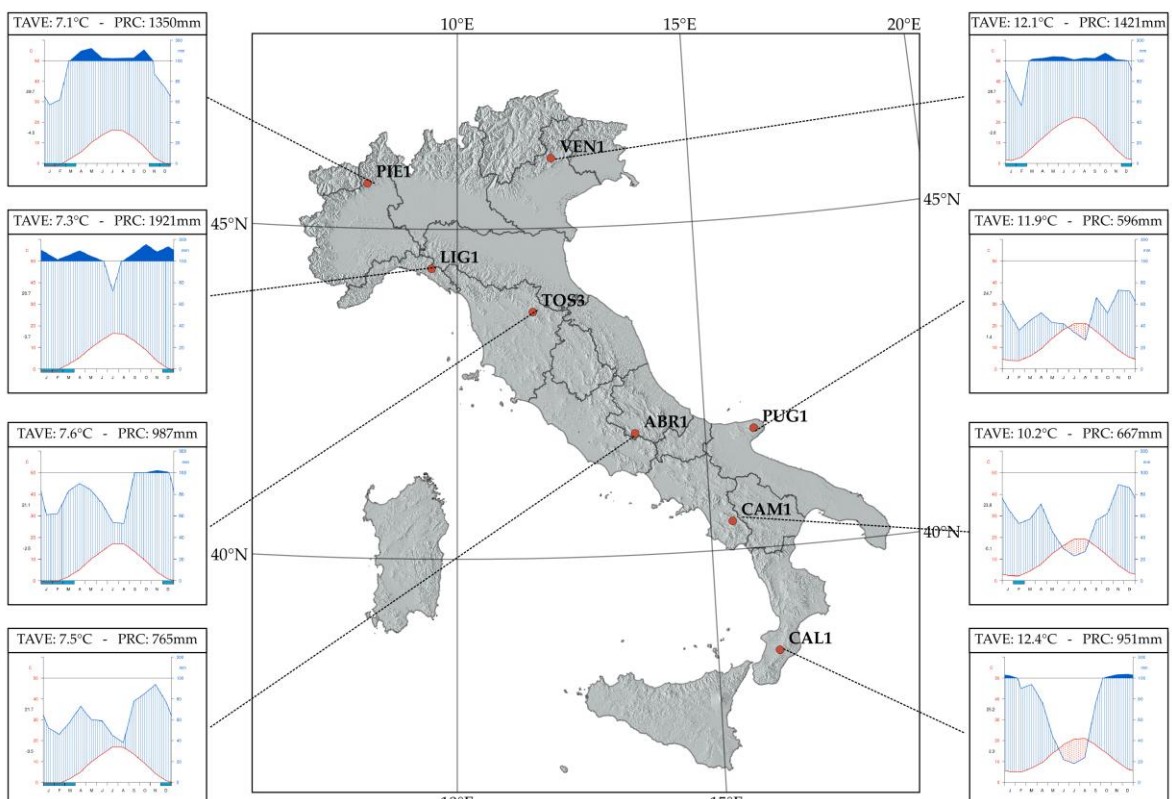

**Figure 1.** Spatial distribution of beech ICP-Forests plots in Italy and Walter &Liethclimatic diagrams for each study site. Climatology was elaborated as the average value of the last available normal climatic period (1981–2010). Average annual temperature (TAVE) and total annual precipitation (PRC) are written at the top of each diagram.

Most plots are grown-up high forest stands. Main mensurational variables per plot are reported in Table 1 and referred to the last survey implemented in 2015.

**Table 1.** Main mensurational variables of beech plots at the last survey (2015) in Italy.

| ICP-Forests Plot | Stand Age (Years) | Total Standing Volume m³ha⁻¹ | Trees Per Hectare (N) | Mean dbh (cm) | Mean Height (m) | Proportion of Beech Trees in Volume % |
|---|---|---|---|---|---|---|
| PIE1 | 80 | 373 | 1109 | 20.1 | 18.5 | 100% |
| VEN1 | 140 | 659 | 337 | 39.1 | 29.6 | 100% |
| LIG1 | 120 | 457 | 336 | 35 | 25.8 | 100% |
| TOS3 | 165 | 981 | 367 | 43.6 | 32.9 | 100% |
| ABR1 | 130 | 630 | 926 | 26.2 | 22.9 | 100% |
| PUG1 | 95 | 755 | 718 | 36.1 | 26.9 | 96.4% |
| CAM1 | 120 | 948 | 236 | 54.6 | 30.6 | 100% |
| CAL1 | 130 | 795 | 251 | 48.3 | 31.4 | 98.6% |

*2.2. Dendrometers and Climate Data*

Inter-annual and tree-level growth dynamics were investigated in this study. Data were collected between 2009 and 2013 from manual UMS-dendrometers installed at the reference tree height of 1.3 m on 15 dominant trees at each plot. Growth course was recorded every 15 to 30 days (Table 2).

**Table 2.** Summary statistics of diameter at breast height (cm) from dendrometers over the 2009–2013 time period, by site.

| Statistic | ABR1 | CAL1 | CAM1 | LIG1 | PIE1 | PUG1 | TOS3 | VEN1 |
|---|---|---|---|---|---|---|---|---|
| Mean | 39.80 | 55.53 | 54.95 | 46.18 | 28.04 | 44.28 | 52.81 | 47.13 |
| StdDev | 0.32 | 0.47 | 0.29 | 0.42 | 0.19 | 0.51 | 0.22 | 0.32 |
| Median | 39.80 | 55.53 | 55.03 | 46.23 | 28.05 | 44.36 | 52.85 | 47.18 |
| Min | 39.26 | 54.79 | 54.43 | 45.49 | 27.72 | 43.29 | 52.47 | 46.58 |
| Max | 40.28 | 56.40 | 55.36 | 46.78 | 28.34 | 45.07 | 53.15 | 47.62 |
| Coef. of Variation | 0.79 | 0.84 | 0.54 | 0.92 | 0.67 | 1.15 | 0.42 | 0.68 |

Growth rates were related to local climate variables to derive possible insights on growth patterns. Even if a meteorological station was available at each ICP-Forest Level II plot, a high amount of missing data was detected in the time series. For this reason, the quality of local climatic time series was lower than the minimum requirements acknowledged for forest monitoring issues [45]. To adopt a common approach for all the study sites, a widely-established, international data source for climate assessment was considered. Monthly minimum and maximum temperature ($T_{mn}$, $T_{mx}$) and total precipitation (P) data were derived from the KNMI (Royal Netherlands Meteorological Institute) Climate Explorer website (https://climexp.knmi.nl/). Summary statistics of climatic data per site are reported in Table 3. Time series of climate variables over the period per site are shown in Figure 2. A Mann-Kendall test was also carried out to detect trends in the monthly data, revealing that no significant trend was detected in climate data, with the exception of precipitation related with the CAM1 site, that showed a significant decrease over the observed time-span.

**Table 3.** Summary statistics of the monthly climatic variables (2009–2013).

| Variable | Statistics | ABR1 | CAL1 | CAM1 | LIG1 | PIE1 | PUG1 | TOS3 | VEN1 |
|---|---|---|---|---|---|---|---|---|---|
| | Mean | 21.18 | 21.18 | 18.46 | 16.48 | 14.76 | 19.89 | 13.54 | 7.77 |
| | StdDev | 6.90 | 6.37 | 6.75 | 7.52 | 7.95 | 7.59 | 7.67 | 7.13 |
| Maximum air temperature (°C) | Median | 20.50 | 20.40 | 17.85 | 16.55 | 15.25 | 18.95 | 13.00 | 7.95 |
| | Min | 10.40 | 11.40 | 7.70 | 4.90 | 1.70 | 7.70 | 1.70 | −4.90 |
| | Max | 32.60 | 32.00 | 29.50 | 28.40 | 27.30 | 32.10 | 26.00 | 19.10 |
| | CV | 32.56 | 30.09 | 36.57 | 45.61 | 53.84 | 38.14 | 56.66 | 91.78 |
| | Mean | 13.52 | 14.89 | 11.51 | 8.95 | 6.06 | 11.15 | 7.13 | 0.59 |
| | StdDev | 5.65 | 5.39 | 5.60 | 6.32 | 6.74 | 6.12 | 6.06 | 6.45 |
| Minimum air temperature (°C) | Median | 13.40 | 14.80 | 11.40 | 8.90 | 6.60 | 10.85 | 6.65 | 0.75 |
| | Min | 3.40 | 5.80 | 1.70 | −2.30 | −5.70 | 0.40 | −3.50 | −11.40 |
| | Max | 23.00 | 23.70 | 20.70 | 19.10 | 16.40 | 21.20 | 16.70 | 10.20 |
| | CV | 41.78 | 36.19 | 48.65 | 70.66 | 111.37 | 54.90 | 85.02 | 1086.27 |
| | Mean | 45.04 | 69.79 | 59.02 | 107.85 | 110.51 | 56.17 | 60.20 | 126.62 |
| | StdDev | 31.25 | 62.14 | 45.38 | 109.89 | 53.41 | 30.43 | 34.84 | 49.19 |
| Monthly precipitation (mm) | Median | 42.85 | 67.15 | 52.00 | 71.45 | 110.35 | 56.85 | 57.30 | 133.95 |
| | Min | 0.10 | 0.00 | 3.20 | 13.30 | 13.10 | 9.60 | 10.70 | 4.20 |
| | Max | 140.40 | 250.80 | 218.00 | 477.70 | 284.10 | 156.00 | 182.80 | 229.70 |
| | CV | 69.37 | 89.03 | 76.89 | 101.89 | 48.33 | 54.18 | 57.87 | 38.85 |

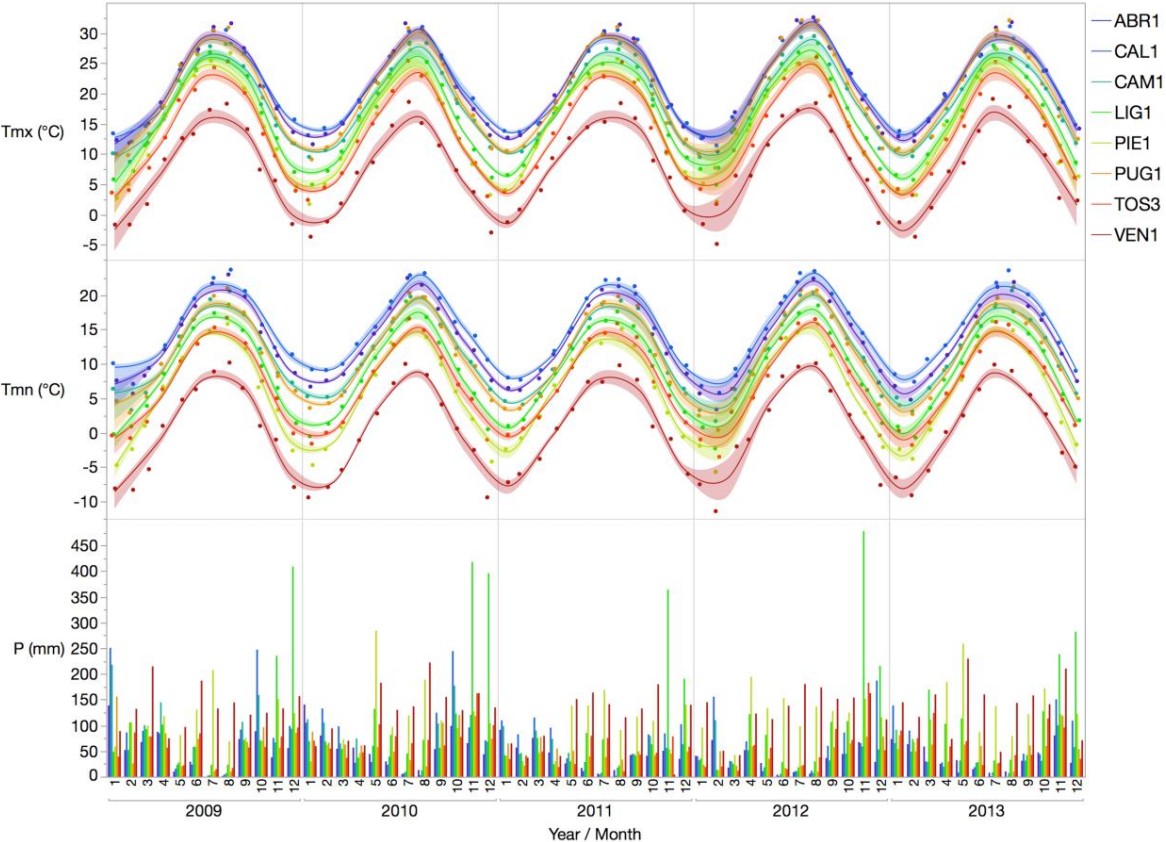

**Figure 2.** Monthly variation of Minimum air temperature (Tmn), Maximum air temperature (Tmx) and Precipitation (P), by site.

### 2.3. Statistical Analysis and Modelling Method

A multivariate approach was used to identify latent patterns among tree growth and climate conditions at the different sites. Firstly, a cluster analysis was performed to explore whether or not similar forest variables could be reduced, and thus represented by the most significant variables in each cluster. An integrated analysis of climatic and dendrometers' data has been performed to explore their possible association and identify latent patterns among climatic data and tree growth. The relationship between dendrometers and climate data was then evaluated by a Principal Component Analysis (PCA), using the different sites as supplementary variables. In this way, sites do not influence the results (eigenvectors/eigenvalues, squared cosines and hence loadings).

Generalized Additive Mixed Models (GAMM) have been implemented (i) to study the influence of the main climatic variables on beech growth trends, (ii) to evaluate whether ecological conditions affected local growth, and (iii) to assess if some plots could be acknowledged as potential sources of adaptation to climatic drivers (i.e., growth not significantly affected by climatic variables). This modelling technique has been recently used to investigate a bimodal growth pattern in Spain [17] and regeneration processes in national forest inventory plots in Germany [46]. GAMMs are an extension of generalized additive models (GAMs)allowing a flexible modelling of nonlinear patterns in the response variable that uses a sum of smoothing functions of the involved covariates and the random variable [47,48].Withthe smooth terms being extremely adaptable, the model does not need to be specified in terms of detailed parametric relationships. For the mixed model fitting the gamm4 [49] package available for R statistical language [50] has been used. Given the structure of the dataset, no data transformation was introduced respecting the natural structure of the phenomena [15] and GAMMs were then used to fit tree-level inter-annual growth, with tree identity as a random factor. Firstly, raw diametric measures were converted into basal area values. Then, the relative increment from the previous survey in a given day of sampling ($G_{inc}$) was calculated as a difference. With this dataset, the GAMM model was built as follow:

$$G_{inc} = s(DOY) + s(DAY) + s(SPEI) + s(T_{mx}) + s(T_{mn}) + s(P) + r \tag{1}$$

The fixed effects in this model are the consecutive day of the year (DOY) ranging between 1 and 365, and expressing the cyclic and periodic change of the seasons, the monitoring day of the analyzed time-span (DAY), expressing the general growth trend over the five years and ranging between 1 (01/01/2009) and 1825 (31/12/2013), the Standardized Precipitation-Evapotranspiration Index (SPEI), the maximum temperature ($T_{mx}$), the minimum temperature ($T_{mn}$) and the precipitation (P) of the reference period of sampling. The random effect denoted by *r*. SPEI variable was obtained from raw temperature and precipitation data using the SPEI package [51] available for R statistical language [50]. The SPEI was calculated with monthly resolution using the Hargreaves equation for reference evapotranspiration (ET0):

$$ET0 = 0.0023 \cdot RA \cdot \left( \frac{T_{mx} + T_{mn}}{2} + 17.8 \right) \cdot \sqrt{T_{mx} - T_{mn}} \tag{2}$$

With RA as the extraterrestrial radiation which is expressed in $MJ.mm^{-1}.day^{-1}$ and all the remaining variables in Celsius degrees. Temperature, precipitation and SPEI variables were used in the GAMM to test the statistical importance of each raw variable ($T_{mx}$, $T_{mn}$, P) and their combined effect (SPEI).

With GAMM, climate-growth relationships were modelled with a nonlinear spline-based smoother coefficient applied to predictors assuming a nonlinear relation between the dependent variable ($G_{inc}$) and the selected ecological predictors. Then all smoothers were tested for statistical significance.

Finally, different climate change scenarios were applied to predict future development at each site. Such scenarios were referred to RCP4.5 and RCP8.5 from the Fifth Assessment Report (AR5) for the 2050s time span. Anomalies were locally downscaled and then added to the current climatic condition. Then, a hierarchical clustering method was performed on the Euclidean distance matrix decomposed using the centroid agglomeration rule in order to assess climatic closeness among the considered sites

as well as to suppose climatic trajectories. To achieve this, 36 widely used climatic variables were derived from raw data. Such climatic variables and indices were firstly proposed by Wang et al. [52] for North America and include many economically or biologically relevant variables such as growing and chilling degree days, heating and cooling degree days, Hargreaves' moisture deficit and reference evaporation. The full list of the 36 variables is reported in Table 4 where each variable was calculated according to Wang et al. [52]. Such variables have often been used in ecological modelling, with the aim to characterize the ecological niche of forest tree species as well as to apply climate change scenarios [53–56] and demonstrated suitability for predicting local climate conditions. Even if directly available from the ClimateEU software (https://sites.ualberta.ca/~{}ahamann/data/climateeu.html), the same data source we used for GAMM (KNMI Climate Explorer website) was here considered to keep the two methods consistent. To avoid collinearity effects in the clustering procedure, the 36 climatic variables and indices were included into a Principal Component Analysis (PCA) centering and scaling the dataset to avoid mensurational biases from different climatic variables (temperatures, precipitation, potential evapotranspiration indices etc.). Then, the 36 components were used as data source for the hierarchical clustering.

**Table 4.** Full list of the 36 climatic variables calculated for the hierarchical clustering and future scenarios forecasting step. All variableswere calculated according to Wang et al. [52] from raw values obtained by KNMI Climate Explorer and used as predictors in the Generalized Additive Mixed Models (GAMM).

| Acronym Used in Recent Literature | Description |
| --- | --- |
| MAT | mean annual temperature (°C) |
| MWMT | mean warmest month temperature (°C) |
| MCMT | mean coldest month temperature (°C) |
| TD | temperature difference between MWMT and MCMT, or continentality (°C) |
| MAP | mean annual precipitation (mm) |
| MSP | mean summer (May to Sept.) precipitation (mm) |
| AH:M | annual heat moisture index (MAT+10)/(MAP/1000)) |
| SH:M | summer heat moisture index ((MWMT)/(MSP/1000)) |
| DD<0 | degree-days below 0°C, chilling degree-days |
| DD>5 | degree-days above 5°C, growing degree-days |
| DD<18 | degree-days below 18°C, heating degree-days |
| DD>18 | degree-days above 18°C, cooling degree-days |
| NFFD | the number of frost-free days |
| FFP | frost-free period |
| bFFP | the Julian date on which FFP begins |
| eFFP | the Julian date on which FFP ends |
| PAS | precipitation as snow between August in previous year and July in current year |
| EMT | extreme minimum temperature over 30 years |
| Eref | Hargreaves reference evaporation |
| CMD | Hargreaves climatic moisture deficit |
| TAV_wt | winter (Dec.(prev. year)–Feb.) mean temperature (°C) |
| TAV_sp | spring (Mar.–May) mean temperature (°C) |
| TAV_sm | summer (Jun.–Aug.) mean temperature (°C) |
| TAV_at | autumn (Sep.–Nov.) mean temperature (°C) |
| TMAX_wt | winter mean maximum temperature (°C) |
| TMAX_sp | spring mean maximum temperature (°C) |
| TMAX_sm | summer mean maximum temperature (°C) |
| TMAX_at | autumn mean maximum temperature (°C) |
| TMIN_wt | winter mean minimum temperature (°C) |
| TMIN_sp | spring mean minimum temperature (°C) |
| TMIN_sm | summer mean minimum temperature (°C) |
| TMIN_at | autumn mean minimum temperature (°C) |
| PPT_wt | winter precipitation (mm) |
| PPT_sp | spring precipitation (mm) |
| PPT_sm | summer precipitation (mm) |
| PPT_at | autumn precipitation (mm) |

## 3. Results

### 3.1. Integrated Analysis of Climatic and Dendrometers' Data

By using a variable clustering method, the dendrometers' data can be represented identifying the most representative variables in each cluster in order to reduce collinearity. Table 5 shows that the cluster-based representative variables are the basal area increment ($G_{inc}$) and the total basal area of the plot (G), explaining almost 100% and 99% of the two clusters' variation as for forest attributes, respectively.

**Table 5.** Cluster summary, reporting the number of variables in each cluster and the variation explained by cluster component.

| Cluster | Number of Members | Most Representative Variable | Cluster Proportion of Variation Explained | Total Proportion of Variation Explained |
|---------|-------------------|------------------------------|-------------------------------------------|------------------------------------------|
| 1 | 2 | $G_{inc}$ | 1 | 0.50 |
| 2 | 2 | G | 0.99 | 0.49 |

The PCA extracted two components explaining about 70% of the total variance. The importance of a component for a given variable can be shown by the squared cosine values (Figure 3a).

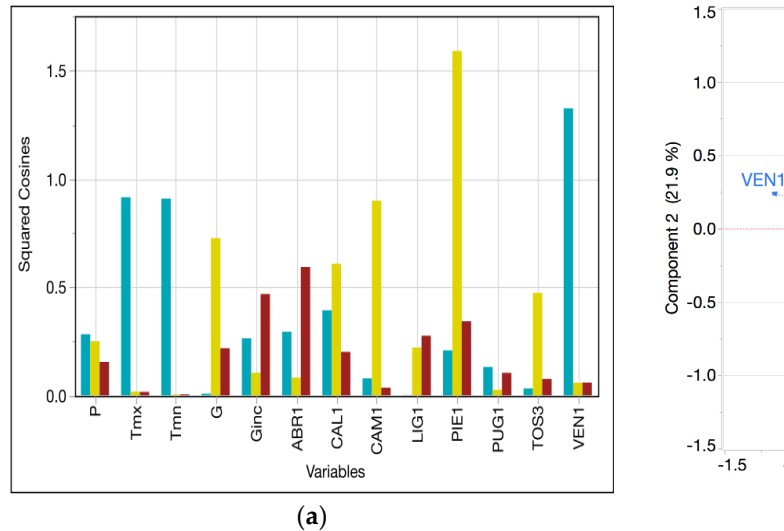 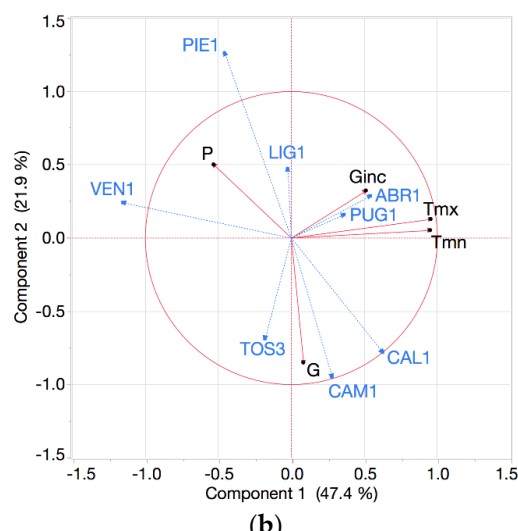

(**a**)          (**b**)

**Figure 3.** Results of a Principal Component Analysis. (**a**) variables squared cosines for the first two principal components. (**a**: climatic and dendrometers' variables; **b**: sites). (**b**) variable loadings for the first two principal components. Supplementary variables are shown in blue and dashed lines.

Variables loadings on the two components, which represent pair-wise correlations between components and variables, are shown in Figure 3, right panel. The analysis has identified changes in the multivariate relationship among the considered variables. Component 1 is positively correlated with maximum and minimum temperature and with PUG1 site and negatively with precipitations and VEN1 site. Component 2 extracts high loadings at the PIE1 site, while it is negatively correlated with G and with CAL1, CAM1 and TOS3 sites. Component 1 and 2 separate the climatic variables (which are mostly represented by Component 1) from the dendrometers' variables, mostly represented by Component 2.

### 3.2. Data Modelling with GAMM

Dendrometers' monitoring provided a high temporal resolution in the assessment of stem seasonal dynamics, as well as tree growth response to environmental variations. The GAMM was always

statistically significant ($p < 0.001$) and explained a relatively low proportion of variance. While a value around 72.94% was shown for LIG1, 68.70% for CAM1, 49.62% for VEN1 and 47.51% for ABR1only 30%–35% of variance was explained at all the other case-studies, leaving much of the variability of $G_{inc}$ unexplained by climatic variables. The most relevant climatic driver was precipitation (P), being statistically significant in all areas, and detected as the most relevant driver also for SPEI which was statistically significant too. Concerning temperatures, both minimum and maximum regimes were highly significant at almost all sites with few exceptions: while maximum temperatures were less significant in PUG1 ($0.05 \leq p$), VEN1 and ABR1 ($0.01 \leq p$), the minimum temperature was poorly significant for growth only in ABR1 ($0.01 \leq p < 0.05$). Then both variables were not significant at all in TOS3 ($p > 0.05$). Finally, as for the cyclic predictors (day of the year, cycling from 1 to 365, DOY), and the variable explaining the monitoring day within the analyzed time span (DAY), while the first was poorly significant in CAM1 only, the DAY was less significant in CAM1, PUG1 and VEN1 and becomes not significant in PIE1 and TOS3.

A summary table of the statistical analysis on growth time series is reported in Table 6.

**Table 6.** Statistical significance of predictors included in the GAMM used to fit tree-level inter-annual growth in relation to the climatic variables with tree identity as a random factor. The statistical significance was tested according to the following thresholds: \*\*\* $p < 0.001$; \*\* $p < 0.01$; \* $p < 0.05$; ns $p \geq 0.05$.

| Variable | ABR1 | CAL1 | CAM1 | LIG1 | PIE1 | PUG1 | TOS3 | VEN1 |
|---|---|---|---|---|---|---|---|---|
| Day of the year (DOY) | \*\*\* | \*\*\* | \* | \*\*\* | \*\*\* | \*\*\* | \*\*\* | \*\*\* |
| Monitoring day (DAY) | \*\*\* | \*\*\* | \*\* | \*\*\* | ns | \*\* | ns | \*\* |
| SPEI | \*\*\* | \*\*\* | \*\*\* | \*\*\* | \* | \*\*\* | \* | \*\*\* |
| Maximum temperature ($T_{mx}$) | \*\* | \*\*\* | \*\*\* | \*\*\* | \*\*\* | \* | ns | \*\* |
| Minimum temperature ($T_{mn}$) | \*\* | \*\*\* | \*\*\* | \*\*\* | \*\*\* | \*\*\* | ns | \*\*\* |
| Total annualprecipitation (P) | \*\*\* | \* | \*\* | \*\*\* | \*\* | \*\* | \*\*\* | \*\*\* |
| Varianceexplained by GAMM % | 47.15 | 36.42 | 68.70 | 72.94 | 29.96 | 42.72 | 37.26 | 49.62 |

*3.3. Data Forecasting*

Hierarchical clustering on future projections was used to assess climatic closeness among the sites as well as to forecast climatic trajectories. The analysis is shown in Figure 4 where two main groups can be detected, based on an integrated latitudinal-longitudinal gradient: the first cluster (upper part of the dendrogram) includes sites from north-western and central Italy relatively close to the Tyrrhenian Sea. The second cluster (lower part of the dendrogram), includes sites from Southern Italy and from a restricted number of sites in central and northern Italy close to the Adriatic Sea. These results may be justified with the different meso-scale climatic regime characterizing the Western (Tyrrhenian) and Eastern (Adriatic) sides of Italy, irrespective of the elevation gradient. As overall, many central and northern monitoring plots are predicted to face climatic conditions very close to those currently detected at southern latitudes such as, for instance, ABR1 and CAM1, which in turn, were predicted to be close to current CAL1 and PUG1 climatic regimes. A special case study was represented by VEN1, whose future and current climates were grouped in a unique and separated cluster.

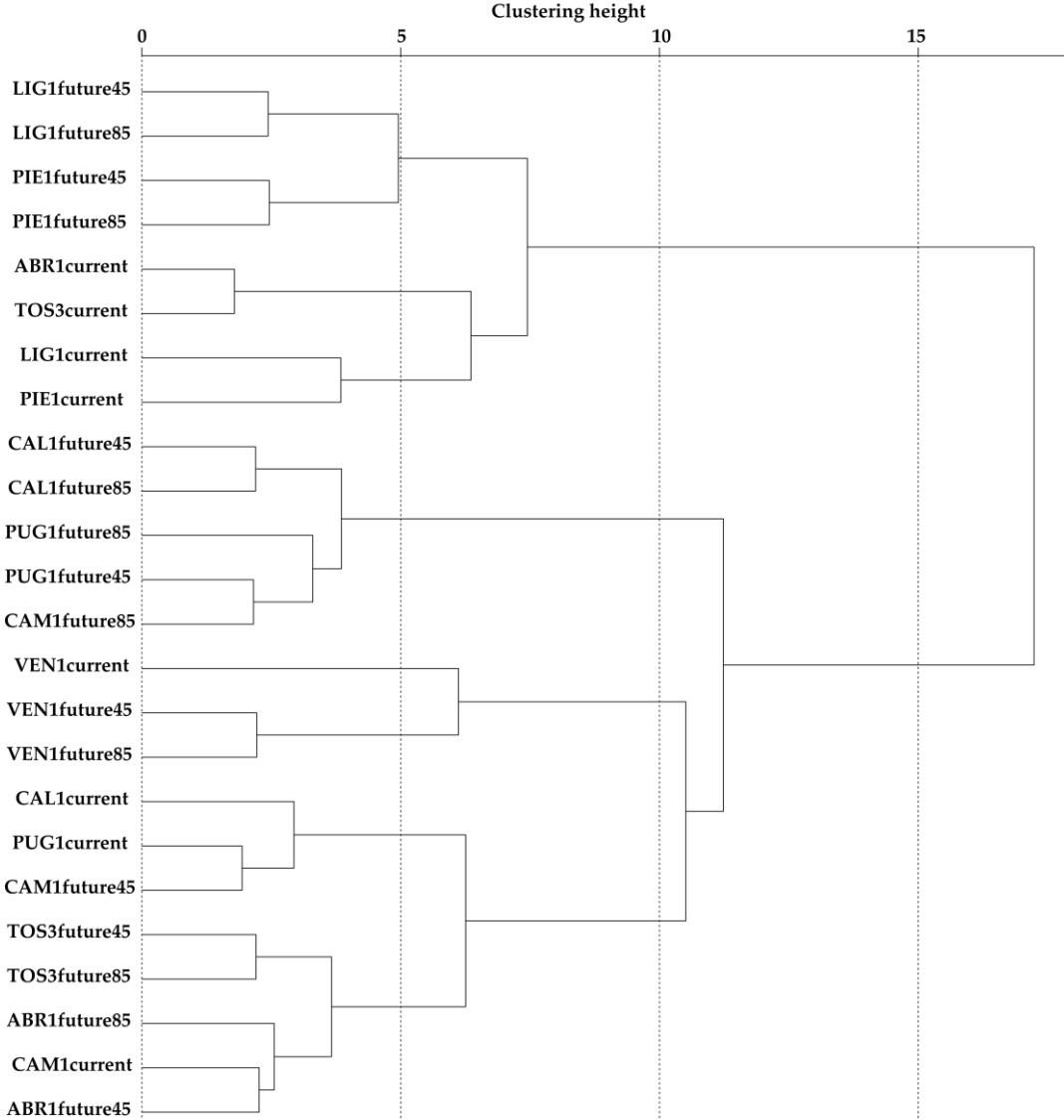

**Figure 4.** Hierarchical clustering for the experimental sites using Euclidean distance matrix decomposed with Ward's agglomeration rule, for different climate scenarios.

## 4. Discussion

The spatial arrangement and growth trend of forest systems are strictly related to management, which influences tree density and vertical/horizontal stand structure. In such a context, GAMM analysis showed how climate was able to explain just a reduced amount of variability across plots. In this case, forest management, i.e., spatial structure and standing volume (not included in our model) might be looked at as the basic drivers for the growth of the analyzed *Fagus sylvatica* trees across Italy.

In a changing environment, accounting for forest response to climate changes is functional to sustainable forest management. Time series represent the time-evolution of the meteorological dynamic process and are fundamental to evaluate patterns and responses of forest species to climate change [26]. The repeated, short-term occurrence of anomalous seasons or extreme events make each year potentially sensitive to climate deviations and therefore even limited time series may provide useful insights today. Many environmental and modelling studies are focused on long-term climatic averages to determine the effects of climate change on forest populations. However, in cases of databases with detailed spatio-temporal resolution, the presence of missing values in a dataset can heavily affect any kind of analysis, leading to the underestimation of natural processes [45]. An increasing degree of

representativeness and precision in statistical techniques to data analysis is needed in studies requiring a higher temporal-resolution, such as dendrochronology, or studies based on seasonal effects on plant growth [17].

Results highlighted latent growth patterns in beech populations along the latitude gradient in Italy, providing an example of the possible use of joint mensurational and climatic high-resolution time series. However, the low variance explained by GAMM was an expected result of this study. Indeed, while most of the variability on growth and stand structure is often driven by climate in non-disturbed stands [9,20,57,58] the same does not hold true for the ICP-Forests network. The analyzed dataset is composed by different forest categories including both coppices (TOS3, LIG1), grown-up high forests (ABR1, CAL1, CAM1, PIE1, PUG1, VEN1) with different structures and responses driven by age [32]. In light of this, our results should be interpreted as a starting point for further investigation. More informative and complete models could actually be built with ICP-Forests datasets including, for instance, soil variables or structural indices which might improve the amount of variance explained in GAMM models. However, we must keep in mind thatthis network is mainly made by managed forest stands, chosen by experts in order to monitor stands' dynamics across a long monitoring period in specific environmental contexts. Consequently, results should be evaluated in the framework of the ICP-Forests only and statistical inference would be just speculative.

Specific outcomes of this study are particularly interesting in light of forest ecology. For instance, the high variance explained by GAMM models in LIG1 might be motivated with the particular environmental condition of the area. In actuality, LIG1 is the first Italian ICP-Forests plot (from North to South) with a small reduction of rainfall over the growing season. The high water availability in a warmer condition than areas located at higher latitudes might influence the growing period without summer stress and a relatively stable climate. Interestingly, the spatially closer plot TOS3 is characterized by different predictors (only SPEI and P were significant) and a peculiar climatic diagram (see Figure 1). The spatial proximity between these two sites, in combination with the results of our study, pointed out the inherent environmental and biological variability across the Mediterranean basin even at small spatial scales and with an (almost) genetically stable forest tree species like European beech [59,60]. Accounting for forest response to climate shift is functional to an adaptive forest management in a changing environment. In this regard, high temporal resolution long-term time series might reveal growth patterns which allow identification of dynamic meteorological-growth relationships [54]. This information is a basic requirement for the ICP-Forests framework, where biotic and abiotic disturbances are being monitored. Moreover, high-quality and long-term time series are increasingly needed for representing meteorological-growth relationship dynamics and fundamental for the evaluation of both processes and responses of forest species to a shifting climate. Even if this issue can be seen as a weakness of this paper, useful information could be read from even a 5-year time series which could drive future monitoring efforts. It is well known that extreme events have been shown to have a crucial role for forest systems and plant communities in the Mediterranean areas [3]. Under future climate changes, such events will play an even more critical role. While the proposed approach can only indirectly outline the importance of climatic extremes in beech forest dynamics, unpredictable events will require longer time-series analysis and refined statistical modelling to understand their effects on forest ecosystems [27]. However, GAMM and dendrometers could be seen as valuable tools for the purpose.

Concerning future scenarios, VEN1 plot represents a peculiar case-study where the current climate and future scenarios seemed to be totally disconnected from all the other beech plots of the Italian monitoring network. Here, local climate regimes might be seen as an important driver to be monitored carefully in the future. An adaptive response of functional traits at the leading and trailing edge of the current distribution will help researchers in understanding possible adaptive measures to balance forest management strategies. Indeed, the central-peripheral adaptive trend will probably play a key role in the framework of a changing climate as an adaptive response of functional traits of small populations, i.e., the marginal and peripheral forest populations [59–61].

Moreover, the bimodal growth of some Mediterranean forest tree species has the potential to affect dendrochronological analysis based on annual tree ring width [17,20]. Knowledge of long-term climate-growth relationships is, therefore, a basic step to address suited future forest management strategies and tackle effectively both climate change-induced effects and human-related disturbances [62,63]. In this respect, a large-scale, intensive monitoring network may provide valuable information about tree growth responses across the annual variability of climate at both latitudinal and elevation gradients.

## 5. Conclusions

The European beech is one of the most important forest tree species in Europe and one of the fundamental patches of the European forest mosaic in both pure and mixed stands. In such a framework, a monitoring effort based on robust statistical models and reliable field data maycontribute information and building up the toolboxin view of the forecasted climate change, includingtree breeding and extensive genotyping as well asadaptive forest management and assisted migration strategies.

**Author Contributions:** Conceptualization, C.F., M.M. and L.S.; methodology, C.F. and M.M.; software, C.F., M.M. and M.P.; validation, G.B. and M.P.; formal analysis, C.F., M.M. and L.S.; investigation, C.F., M.M., S.F. and L.S.; resources, G.F. and S.F.; data curation, M.P. and G.B.; writing—original draft preparation, C.F., M.M. and G.F.; writing—review and editing, C.F., M.M., G.B., S.F. and L.S.; visualization, M.P.; supervision, G.F., S.F. and L.S.; project administration, S.F. and G.F.; funding acquisition, S.F., G.F. and L.S.

**Funding:** The research was funded by the Smart4Action LIFE + project "Sustainable monitoring and reporting to inform forest- and environmental awareness and protection" (LIFE13 ENV/IT/000813).

**Acknowledgments:** Thank you to Tiziano Sorgi and Valerio Moretti for their helpful technical assistance. Authors wish to thank the two anonymous reviewers, for their professional and high-quality work in improving this manuscript during the peer-review process.

**Conflicts of Interest:** The authors declare no conflict of interest.

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
