# Peer review of "Exploring Nonlinear Intra-Annual Growth Dynamics in Fagus sylvatica L. Trees at the Italian ICP-Forests Level II Network"

_forests, doi:10.3390/f10070584_

Round 1

Reviewer 1 Report

Review
  The work presented for the review concerns the climate-beech relation in Italy. Overall, I found that the work is interesting, the presented research material is very valuable. However, the work requires supplementing the content and some corrections.

Main aim of the work ……………………..,to evaluate the intra-annual growth rate of European beech and identify possible forest management strategies to be adopted in the framework of the ICP-Forest network.

In presented work, unfortunately, I didn’t find clear responses to main aims of the research, especially about strategies. 

In my opinion, the influence of the main climatic variables on beech growth trends on the basis of 5-years measurements can be affected by error. 

Detailed comments:

1.       Introduction

 Line 104- 107 - In my opinion, the influence of the main climatic variables on beech growth trends on the basis of 5-years measurements can be affected by error. 

2.       Material & methods

Figure -Information on climate diagrams, for example, T mean and P is not clearly visible (maybe bigger print).

Line 154 - reference to table 2 is unnecessary

Comment to Table 3 - in climatological studies, the average monthly precipitation is not presented in the way as in Table 3. This is due to the high variability of rainfall from month to month, which the authors indicated in Figure 2 and in the table. Maybe annual total of precipitation or precipitation during in growing season or precipitation from the period according to the results of dendrochronological research [see e.g. Piovesan et all 2003]

Maybe some information about model GAMM fitting

3.       Results

3.1

 Figure 3.  There is lack of explanation of variable G and bars in Figure 3 left

 In Table 4 there is lack representative variables:  M3A and TNL, there are variables Ginc i G

3.3 Data forecasting

What input data were used in climatic scenarios, what was the time range of those data?

4.        Conclusion

Conclusion seems to be modest considering, I didn’t find clear responses to main aims of the research, especially about strategies

Author Response

@page { margin: 0.79in } p { margin-bottom: 0.1in; direction: ltr; line-height: 115%; text-align: left; orphans: 2; widows: 2 } a:link { so-language: zxx }

Reviewer#1

Comments and Suggestions for Authors

The work presented for the review concerns the climate-beech relation in Italy. Overall, I found that the work is interesting, the presented research material is very valuable. However, the work requires supplementing the content and some corrections.

Main aim of the work ……………………..,to evaluate the intra-annual growth rate of European beech and identify possible forest management strategies to be adopted in the framework of the ICP-Forest network. In presented work, unfortunately, I didn’t find clear responses to main aims of the research, especially about strategies.

Authors: The text has been adjusted according to the reviewer’s comment, writing that the aim is to identify possible climate-growth interactions

In my opinion, the influence of the main climatic variables on beech growth trends on the basis of 5-years measurements can be affected by error.

Authors: The reviewer (as also highlighted by a second reader, Rev#2) is right in regard with the relatively small time period we considered. However, as stated in the title with the verb “exploring”, this paper should be seen as the first step of a longer project aimed at “predicting” climate change effects on Fagus sylvaticatrees and where (we do hope) in future time series will be longer and more robust. The mixed approach proposed in this paper, integrating multivariate statistics with formal inferential models like a GAMM, is appropriate to reduce errors and redundancy among relevant variables.

Detailed comments:

1. Introduction

Line 104- 107 - In my opinion, the influence of the main climatic variables on beech growth trends on the basis of 5-years measurements can be affected by error

Authors: please see the same answer as above

2. Material & methods

Figure -Information on climate diagrams, for example, T mean and P is not clearly visible (maybe bigger print).

Authors: The figure has been modified

Line 154 - reference to table 2 is unnecessary

Authors: The reference was removed

Comment to Table 3 - in climatological studies, the average monthly precipitation is not presented in the way as in Table 3. This is due to the high variability of rainfall from month to month, which the authors indicated in Figure 2 and in the table. May be annual total of precipitation or precipitation during in growing season or precipitation from the period according to the results of dendrochronological research [see e.g. Piovesan et al. 2003]

Authors: we believe the reviewer stressed an interesting point. However, we decided to keep this table and figure and to add the average annual temperature and the total annual precipitation as text in Figure 1. We did this because we prefer to use this table as a description of the dataset we used for modelling more than the data to characterise the ICP-Forest plots

Maybe some information about model GAMM fitting

Authors: additional information were added in section 2.3

3. Results

3.1

Figure 3.  There is lack of explanation of variable G and bars in Figure 3 left

Authors: A specific explanation has been added

In Table 4 there is lack representative variables:  M3A and TNL, there are variables G inc i G

Authors: We have corrected the text accordingly

3.3 Data forecasting

What input data were used in climatic scenarios, what was the time range of those data?

Authors: a wider explanation has been added to avoid this shortcoming

4. Conclusions

Conclusion seems to be modest considering, I didn’t find clear responses to main aims of the research, especially about strategies

Authors: the conclusion section has been modified and revised substantially for clarity and brevity.

Reviewer 2 Report

Summary:

While the manuscript is generally well written, I have doubts regarding the model. Maybe most importantly, no interaction between precipitation and temperature, or a drought index like SPEI is considered. Further more the 'DAY' effect in the model is unreasonable as 'age effect'. Forest structure parameters are not considered in the models and only hypothesized to play an important role. While dendrometer data does provide a higher resolution of radial tree growth, the downside of is the low sample size (5 years).

More specific comments:

Five years of tree-growth is of-course a pretty small sample size. This has some implications for the  statistical analyses that you might point out to the readers. For example, the effect size (explained variance) of each environmental variable depends also on the variance of this variance in these 5 years; e.g. if June precipitation in all years was very similar (low variance), then this variable will explain little variance. So either mention this uncertainty, or state/show that the environmental variance in these 5 years was similar to the long-term (several decades) variance.

Dendrometers only on dominant trees: While certain climatic signals might be stronger in dominant trees, only sampling such dominant trees will not be representative for the climate signal of the forest/population (see Nehrbass-Ahles et al., 2014).

GAMM: Why are you using the “DAY” effect in your model? Trees get hundreds of years old, so five years will matter very little. What this variable likely encodes (and why it is sometimes significant in your model) might rather be the general growth trend over these five years. This is not an age effect. Further more, it is tree size rather than age, that affects growth. In stead of using the 'DAY' variable it would be more meaningful to use each tree's basal area (at the respective 'DAY') as input variable? This way you could account for tree size effects within as well as between trees.

Time period: In the model you use data from 01/01/2009 to 31/12/2014, i.e.  2191 days. This, is six(!) years of data, but in the rest of the manuscript you say you have five years of data!? Climate data is only shown until the end of 2013.

Temperature and precipitation interaction: Your model does not account for the fact that precipitation and temperature interact: the same amount of precipitation matters more or less depending on the temperature, because temperatures affects evapotranspiration. You might for example consider calculating a drought index like SPEI or the vapor-pressure deficit. Drought indices could have the advantage that they account for lagged soil moisture effects. For example if there was much rain in May, drought in June matters less.

Modelling the predictors: You write: 'All predictors were modeled with a nonlinear spline-based smoother with the aim to account for a hypothesized nonlinear relation between the dependent variable (Ginc) and selected ecological predictors.' I am not sure if I understand this right and it would be nice if you could clarify this. Do you mean

a)you fitted a spline through your time-series data, so you get from monthly (climate) variables to the daily resolution or

b) do you fit a spline through the relationship between Ginc and the predictor and then use this spline values as model input?

c) ???

Maybe I just do not understand it since I am not too familiar with GAMMs but phrases like 'using a sum of smoothing functions of the involved covariates and the random variable' might be explained better.

If not climate, what drives tree growth?: You hypothesize that forest management trajectories have large effect on tree growth, which also seems likely to me. But why do you not try to quantify this affect? You do have some important meta data like standing volume per hectare, mean height or stand age, which could be useful explanatory variables. This would of course mean using all site's data in one model, not one model per site.

Modelling growth under climate change scenarios: Estimating how trees will grow under different climate change scenarios, based on five years of dendrometer data sounds a bit overambitious to me but you are of course free to do this as long as you discuss the uncertainties. In the end of the abstract (line 55 and further) you also write that northern Italy will in the future face similar climatic conditions as currently in more southern latitudes; but does that also mean that beech will grow as beech is currently growing in more southern latitudes?

Comments regarding the Figurs

Fig 1: The axes of the climate diagrams cannot be read because they are too small. Everything needs to be readable when printed on A4 (100% zoom level).

Fig 2: In line with my comment on the short length of your time-series data, it would be nice to show climate data with something like a 'jitter plot' for monthly variables. You already have the climate data since I assume that climate data could be downloaded for more than just 5 years from this website. So it would be nice to see how for example June temperature was distributed over the past >50 years (jitterplot, violin plot, boxplot etc.) and how June temperature was in your respective 5 years. Until now, this figure only allows to compare climate between sites. Comparing climate between years is not easily possible with this kind of plot.

Comments by line:

44: when three nouns are in a row, two should be connected with a dash: “forest tree-species”

150: MonthLY

157: monthLY

221: what are these 'cluster-based representative variables' M3A and TNL? You never define it and table 4 does not mention these either.

197: the the lower and upper case letter in the formula intentional?: 's' and 'S'

p { margin-bottom: 0.25cm; line-height: 120%; }

Author Response

@page { margin: 0.79in } p { margin-bottom: 0.1in; direction: ltr; line-height: 115%; text-align: left; orphans: 2; widows: 2 } a:link { so-language: zxx }

Reviewer#2

Summary

While the manuscript is generally well written, I have doubts regarding the model. May be most importantly, no interaction between precipitation and temperature, or a drought index like SPEI is considered. Furthermore the 'DAY' effect in the model is unreasonable as 'age effect'. Forest structure parameters are not considered in the models and only hypothesized to play an important role. While dendrometer data does provide a higher resolution of radial tree growth, the downside of is the low sample size (5 years).

Authors: the concerns raised by the reviewers are generally sound and should be kept in a proper attention. GAMM is not able to handle interactions. To solve this, linear (or nonlinear) mixed model should be run but this would reduce the strength of the analysis given the impossibility to use smoothing functions. Concerning SPEI, we worked to include it in the paper. As a result, the new GAMM now included also SPEI as predictor. Please note that also the former predictors were kept this was done to improve the model. Actually using just SPEI as predictor the statistical significance was lower as well as the explained variance. In addition, we can now see which raw climatic variables (Tmin, Tmax, Prec) were mainly responsible for SPEI significance. We hope this new model could answer, at least partially, the point raised here.

Concerning DAY, this value is not between 1 and 365 but it is a continuous number between 1 and  1825 which only increases across the whole study. The parameter cycling from 1 to 365 every year was DOY (Day Of the Year). For this reason, we wrote that DAY was “the monitoring day of the analysed time-span (DAY), expressing the general growth trend over the considered five years ranging between 1 (01/01/2009) and 1825 (31/12/2013)“

More specific comments:

Five years of tree-growth is of-course a pretty small sample size. This has some implications for the statistical analyses that you might point out to the readers. For example, the effect size (explained variance) of each environmental variable depends also on the variance of this variance in these 5 years; e.g. if June precipitation in all years was very similar (low variance), then this variable will explain little variance. So either mention this uncertainty, or state/show that the environmental variance in these 5 years was similar to the long-term (several decades) variance.

Authors: The reviewer (as also highlighted by Rev#1) is right concerning the small time period we considered. However, as stated in the title with the verb “exploring”, this paper should be seen as the first step of a longer project aimed at “predicting” climate change effects on Fagus sylvatica trees and where (we do hope) future time series will be longer and more robust. We expressed in a more specific way, the advantage and limits to evaluate a relatively short time period in terms of variance explained and potential to capture the impact of extreme events. This comment was specifically added to the discussion section. We remain available to elaborate further this point.

Dendrometers only on dominant trees: While certain climatic signals might be stronger in dominant trees, only sampling such dominant trees will not be representative for the climate signal of the forest/population (see Nehrbass - Ahles et al., 2014).

Authors: we believe that the comment of the author is not completely true. As always stressed in dendrochronological studies, the dominant layer of a forest stand is the most sensitive to climate. We are aware that this is NOT the average behaviour of the population BUT it well expresses the sensitivity of the species to climate. In this sense - and the title supports this thought - the paper is aimed at estimating the influence of climate on the species across the selected ICP-Forests plots across Italy, not on the analysed stands neither to make inferences on the species across the whole country. Indeed, given the not probabilistic sampling scheme of the ICP-Forests plots, we are aware that no inferences are allowed.

GAMM: Why are you using the “DAY” effect in your model? Trees get hundreds of years old, so five years will matter very little. What this variable likely encodes (and why it is sometimes significant in your model) might rather be the general growth trend over these five years. This is not an age effect. Furthermore, it is tree size rather than age, that affects growth. Instead of using the 'DAY' variable it would be more meaningful to use each tree's basal area (at the respective 'DAY') as input variable? This way you could account for tree size effects within as well as between trees.

Authors: concerning DAY this value is not between 1 and 365 but it is a continuous number between 1 and 1825 which monotonically increases across the whole study. For this reason DAY was just a proxy of the ageing effect. Then you are right writing that “Trees get hundreds of years old, so five years will matter very little” and we agree that “What this variable likely encodes (and why it is sometimes significant in your model) might rather be the general growth trend over these five years”. This was exactly the hidden meaning we were thinking for our paper. Your description and comments have been incorporated in the paper. Thanks for this suggestion.

Time period: In the model you use data from 01/01/2009 to 31/12/2014, i.e. 2191 days. This, is six(!) years of data, but in the rest of the manuscript you say you have five years of data!? Climate data is only shown until the end of 2013

Authors: this was a typo error. Corrected accordingly.

Temperature and precipitation interaction: Your model does not account for the fact that precipitation and temperature interact: the same amount of precipitation matters more or less depending on the temperature, because temperatures affects evapotranspiration. You might for example consider calculating a drought index like SPEI or the vapor-pressure deficit. Drought indices could have the advantage that they account for lagged soil moisture effects. For example if there was much rain in May, drought in June matters less.

Authors: as we wrote for the previous comment, this would be a very interesting index so we tried to include it in the paper. See now the new model where SPEI was used.

Modelling the predictors: You write: 'All predictors were modelled with a nonlinear spline-based smoother with the aim to account for a hypothesized nonlinear relation between the dependent variable (G inc) and selected ecological predictors.' I am not sure if I understand this right and it would be nice if you could clarify this. Do you mean[...]. Maybe I just do not understand it since I am not too familiar with GAMMs but phrases like 'using a sum of smoothing functions of the involved covariates and the random variable' might be explained better.

Authors: the sentence was unclear so we rephrased it as “Climate-growth relationships were modelled with a non-linear spline-based smoother coefficient applied to predictors with the aim to account for a hypothesized nonlinear relation between the dependent variable (Ginc) and the selected ecological predictors.

If not climate, what drives tree growth? You hypothesize that forest management trajectories have large effect on tree growth, which also seems likely to me. But why do you not try to quantify this affect? You do have some important metadata like standing volume per hectare, mean height or stand age, which could be useful explanatory variables. This would of course mean using all site's data in one model, not one model per site.

Authors: the reviewer is right and the comment is useful and pertinent. However, the low variability of the mensurational variables within the 5 years was not able to explain the intra-annual differences in growth, which is the focal point of this paper (pointed out in the new title). We agree with him/her but a different model would completely change the paper, driving it away from our original aims. Then similar papers have already been written by some of the authors of this paper (see for instance reference [34]) with longer time-series and a different type of dataset. Some parts have been added in the fourth section to better discuss this point.

Modelling growth under climate change scenarios: Estimating how trees will grow under different climate change scenarios, based on five years of dendrometer data sounds a bit overambitious to me but you are of course free to do this as long as you discuss the uncertainties. In the end of the abstract (line 55 and further) you also write that northern Italy will in the future face similar climatic conditions as currently in more southern latitudes; but does that also mean that beech will grow as beech is currently growing in more southern latitudes?

Authors: this is exactly the key point and what monitoring efforts should do: to compare empirical evidence with the modelling results in order to understand whether predictions and models were reliable.

Comments regarding the Figures

Fig 1: The axes of the climate diagrams cannot be read because they are too small. Everything needs to be readable when printed on A4 (100% zoom level).

Authors: the figure has been slightly reshaped also according to comments provided by Rev#1. However, we were not interested in showing exact values of axis while showing graphically how the climatic features can change across the country for the same species.

Fig 2: In line with my comment on the short length of your time-series data, it would be nice to show climate data with something like a 'jitter plot' for monthly variables. You already have the climate data since I assume that climate data could be downloaded for more than just 5 years from this website. So it would be nice to see how for example June temperature was distributed over the past >50 years (jitterplot, violin plot, boxplot etc.) and how June temperature was in your respective 5 years. Until now, this figure only allows to compare climate between sites. Comparing climate between years is not easily possible with this kind of plot.

Authors: we believe the reviewer stressed an interesting point. However, we decided to keep Figure 2, since it is useful to describe the analysed time-period, also considering that Figure 1 already provides the climate general information in Italy over the last years.

Comments by line:

44: when three nouns are in a row, two should be connected with a dash: “forest tree-species” Authors: changed accordingly.

150: MonthLY

Authors: changed

157: monthly

Authors: changed

221: what are these 'cluster-based representative variables' M3A and TNL? You never define it and table 4 does not mention these either.

Authors: We have correct the text and more explanation has been added

197: the lower and upper case letter in the formula intentional?: 's' and 'S'

Authors: no, it was a typo error. Corrected accordingly.

Round 2

Reviewer 1 Report

The manuscript has been significantly improved, majority my comments and suggestions have been included in the revised version. Now the manuscript is well written, but there are few typo errors in the text. e.g line 352, 386.

Author Response

@page { margin: 2cm } p { margin-bottom: 0.25cm; direction: ltr; color: #000000; line-height: 115%; text-align: left; orphans: 2; widows: 2 } p.western { so-language: en-GB } p.cjk { so-language: en-US } p.ctl { so-language: ar-SA } a:link { so-language: zxx }

Reviewer #1

The manuscript has been significantly improved, majority my comments and suggestions have been included in the revised version. Now the manuscript is well written, but there are few typo errors in the text. e.g line 352, 386.

Authors: We thank the reviewer for his helpful comments. We apologise for the typos which probably occurred due to the tack change version and masked by different colours, deleted text, moved sentences and so on. We are now submitting a clean version where such errors should be avoided.

Reviewer 2 Report

The manuscript was improved and most comments from the previous review were incorporated.

Minor language edits could be done by a native speaker (unlike me) to improve the manuscript.

(I only remark on one tiny typo in line 353: "he" should be "the")

Regarding the methodology I would still argue that the 'DAY' variable should be removed from the model. This variable captures a trends over the whole time period and might thus contain parts of the variance that would otherwise be explained by variables like temperature, SPEI or any variable interaction. However, I leave this model-design decision up to the authors.

I am looking forward to the authors next articles when more data becomes available.

Author Response

Reviewer #2

The manuscript was improved and most comments from the previous review were incorporated. Minor language edits could be done by a native speaker (unlike me) to improve the manuscript. (I only remark on one tiny typo in line 353: "he" should be "the")

Authors: As stated for Reviewer #1 we thank he/she for his/her helpful comments. We apologise for the typos which probably occurred due to the tack change version.

Regarding the methodology I would still argue that the 'DAY' variable should be removed from the model. This variable captures a trends over the whole time period and might thus contain parts of the variance that would otherwise be explained by variables like temperature, SPEI or any variable interaction. However, I leave this model-design decision up to the authors.

Authors: We believe your comment is pertinent and timing. However when running GAMM without DAY the explained variance tend to decrease with very low impact on the statistical significance of predictors. That’s why we decided to leave the model as it was in the last submitted paper

I am looking forward to the authors next articles when more data becomes available.

Authors: we are looking forward for the next paper too, waiting data to be delivered by ICP-Forests, hopefully with more climate data too. Thanks again for your helpful comments. We are grateful and satisfied by this final version of our paper, greatly improved also thanks to your comments.

This manuscript is a resubmission of an earlier submission. The following is a list of the peer review reports and author responses from that submission.

Round 1

Reviewer 1 Report

Review
  The paper presented for the review concerns the relation of climate-beech growing in Italy. The paper is interesting, the presented research material is very valuable.
However, the paper requires supplementing the content and some corrections. There are errors related to references from Tables, errors related to the lack of explanations and shortcomings regarding the citation of literature.

 The work requires supplementations with:

-          reference to research connected with relation climate -growth of beech in Italy, studied by dendrochronological methods

-          reference to studies on the impact of climate change on beech growth in Italy

-          more information about Generalized Additive Mixed Models (GAMMs) used at work

Detailed comments:

1.       Introduction
In the literature review there is no information on the climate-growth relation of Fagus sylvatica in Italy. Research on the relation of climate-beech growth has been carried out in Italy on a large scale, both in the altitude and latitudinal gradient, e.g.

       Gianluca Piovesan , Franco Biondi, Mauro Bernabei, Alfredo Di Filippo, Bartolomeo Schirone; Spatial and altitudinal bioclimatic zones of the Italian peninsula identified from a beech (Fagus sylvatica L.) tree-ring network

·         Piovesan, G., Bernabei, M., Di Filippo, A., Romagnoli, M., Schirone, B., 2003. A long-term tree ring beech chronology from a high-elevation old-growth forest of Central Italy. Dendrochronologia 21, 1–10.

·         Piovesan, G., Di Filippo, A.,Alessandrini, A., Biondi, F., Schirone, B., 2005. Structure, dynamics, and dendroecology of an Apennine old-growth beech forest. J. Veg. Sci. 16, 13–28.

·         Di Filippo, Biondi F, Čufar K., Martín De Luis, Michael Grabner, Maurizio Maugeri, Emanuele Presutti Saba, Bartolomeo Schirone, Gianluca Piovesan; Bioclimatology of beech (Fagus sylvatica L.) in the Eastern Alps: spatial and altitudinal climatic signals identified through a treering network

·         Chelli I in. 2017.  Climate change response of vegetation across climatic zones in Italy, Climate Research 71(3):249-262

2.       Material & methods

Figure 1

There is lack of information which years (period) were used to construct climate diagrams.

Was it 5 years in which the study was conducted (2009-2013) or is other period e.g. the latest climatic normal period?
Information on climate diagrams, for example, T mean and P is not clearly visible (maybe bigger print).

Line 115 - incorrect reference to table 2, table 2 applies to climatic variables

Line 123 - incorrect reference to table 3,

Lines 123-124 - in Figure 2 are presented time series (the course of average monthly values in 2009-2013), there is no analysis of the trend of the series.

Comment to Table 2 - in climatological studies, the average monthly precipitation is not presented in the way as  in Table 3. This is due to the high variability of rainfall from month to month, which the authors indicated in Figure 2 and in the table.

Suggestion to table 3 - it would be better to give the average value of dbh increment in 2009-2013 for research plots.

Line 145 – There is lack of information about this modeling method as per quoted literature/ as per cited literature.

Questions and comment referring to GAMM model

1. Which smoother was chosen and why?

2. How the model was reduced from over-fitting?

Comparison of GAM fitting and loess fitting (figure 4) indicates that the model is not well-matched.  It seems to me, that the authors should try to find a better fit of the model. If this model GAM is the best fit, such information should be found at paper.

2. How the model was reduced from over-fitting?

Comparison of GAM fitting and loess fitting (figure 4) indicates that the model is not well-matched.  It seems to me, that the authors should try to find a better fit of the model. If this model GAM is the best fit, such information should be found at paper.

3.  Results

3.1

Lines 172-174 – In Table 4 there is lack representative variables:  M3A and TNL,  there are variables Ginc i G

Figure 3 and line 187 – There is lack of explanation of variable G and bars in Figure 3 left

 3.3 Data forecasting

What input data were used in climatic scenarios, what was the time range of those data?

4.        Discussion

Discussion seems to be modest considering amount of research in relation of climate-growth of beech as well as using models in those research (look po.1).

The main goals of research were: 

 Lines 78-82 and 148-151

 “Main  aim of the work …………………….., is to explore and improve the understanding of (apparent and latent) relationships between environmental factors  (seasonal climate variability) and radial growth pattern (as a response and explanatory variable of tree condition)” .

“In this paper, this modelling technique has been implemented to study the influence of the main climatic parameters on growth trends of the species and to evaluate whether ecological conditions affected local growth and if some plots could be acknowledged as potential source of adaptation to climatic drivers (i.e. growth not significantly affected by this parameter).

In presented paper, unfortunately, I didn’t find clear responses to main aims of the research.  

.The authors conclude the influence of precipitation and thermal conditions, but they do not determine the relationship between the seasonal variability of climatic factors and the radial growth pattern.

Reviewer 2 Report

The topic in general is interesting and the data seems adequate but the research questions for this manuscript are unclear, and the results and discussion presented seem still preliminary and vague. The paper might still require more effort in the statistical analysis, but it definitely needs to be improved with respect to the clarity of presentation of the results and the discussion of the results with the relevant literature. My review term is therefore "reject".